# A Novel *DUF569* Gene Is a Positive Regulator of the Drought Stress Response in Arabidopsis

**DOI:** 10.3390/ijms22105316

**Published:** 2021-05-18

**Authors:** Rizwana Begum Syed Nabi, Rupesh Tayade, Adil Hussain, Arjun Adhikari, In-Jung Lee, Gary J. Loake, Byung-Wook Yun

**Affiliations:** 1School of Applied Biosciences, Kyungpook National University, Daegu 41566, Korea; rizwananabi@korea.kr (R.B.S.N.); rupesh.tayade@gmail.com (R.T.); xteriousarjun7@gmail.com (A.A.); ijlee@knu.ac.kr (I.-J.L.); 2Department of Southern Area Crop Science, National Institute of Crop Science, Rural Development Administration, Miryang 50424, Korea; 3Department of Agriculture, Abdul Wali Khan University, Mardan 230200, Pakistan; adilhussain@awkum.edu.pk; 4Institute of Molecular Plant Sciences, School of Biological Sciences, University of Edinburgh, King’s Buildings, Edinburgh EH9 3JH, UK

**Keywords:** Arabidopsis, drought, *DUF569*, antioxidant activity

## Abstract

In the last two decades, global environmental change has increased abiotic stress on plants and severely affected crops. For example, drought stress is a serious abiotic stress that rapidly and substantially alters the morphological, physiological, and molecular responses of plants. In Arabidopsis, several drought-responsive genes have been identified; however, the underlying molecular mechanism of drought tolerance in plants remains largely unclear. Here, we report that the “domain of unknown function” novel gene *DUF569* (*AT1G69890*) positively regulates drought stress in Arabidopsis. The Arabidopsis loss-of-function mutant atduf569 showed significant sensitivity to drought stress, i.e., severe wilting at the rosette-leaf stage after water was withheld for 3 days. Importantly, the mutant plant did not recover after rewatering, unlike wild-type (WT) plants. In addition, *atduf569* plants showed significantly lower abscisic acid accumulation under optimal and drought-stress conditions, as well as significantly higher electrolyte leakage when compared with WT Col-0 plants. Spectrophotometric analyses also indicated a significantly lower accumulation of polyphenols, flavonoids, carotenoids, and chlorophylls in *atduf569* mutant plants. Overall, our results suggest that novel *DUF569* is a positive regulator of the response to drought in Arabidopsis.

## 1. Introduction

The life cycle and production of plants are strongly affected by exposure to harsh environmental conditions [1]. To protect against such negative conditions, plants have developed various survival strategies by which they adapt to numerous biotic and abiotic stresses [2]. However, abiotic stresses (e.g., drought, salinity, high or low temperature, and heavy metals) are dynamic and complex. To combat abiotic stress, plants have evolved compound mechanisms over time with changes occurring at the physiological, cellular, and transcriptomic levels [3]. (With increasing environmental change, drought and high salinity are the two key abiotic stresses that limit the growth, development, and yield of crop plants impacting food security [4,5,6].

Water shortage leads to the accumulation of reactive oxygen species (ROS) and/or reactive nitrogen species (RNS). Elevated ROS and RNS levels result in nitrooxidative stress that severely affects cell structure and integrity and may lead to, for example, the loss of organelle function, reduced metabolic function, lipid peroxidation, programmed cell death, and electrolyte leakage [7]. In addition, the resultant oxidative stress causes a reduction in photosynthetic rate and electron flow that leads to oxidative damage [8]. To counteract these ROS- and RNS-mediated negative effects, plants have developed sophisticated enzymatic and non-enzymatic mechanisms. Enzymatic mechanisms include catalases (CATs) [9], peroxidases (PODs), and metallothionein ROS scavenging activity, whereas non-enzymatic mechanisms involve vitamins C and E, glutathione, flavonoids, polyamines, and carotenoids. CAT activity has been found to increase during abiotic stress conditions across plant species [10]. In Arabidopsis, the CAT gene family comprises *CAT1*, *CAT2*, and *CAT3*, which encode the CAT proteins. Functional characterization of these genes has revealed that *CAT2* is the most responsive; for example, the loss of function mutant *atcat2* shows significantly reduced CAT activity compared with *CAT1* and *CAT3* mutants [11]. Antioxidant enzymes such as PODs and polyphenol oxidase (PPO) are also involved in oxidative stress regulation. Furthermore, the accumulation of the phytohormone abscisic acid (ABA) occurs in plants under drought stress; this activates NADPH oxidases that produce reactive oxygen intermediates [12,13]. The general role of ABA in plants, which regulates a diverse range of cellular and molecular processes, is well established, but ABA is also involved in regulating the adaptive response to abiotic stresses such as drought and salinity [14]. In addition to antioxidants and phytohormones, the RNS nitric oxide (NO) can play an important role in modulating the response of plants to drought and oxidative stress [15,16]. Indeed, studies have demonstrated that NO plays a key role at the post-transcriptional level to alter the structure and associated function of plant proteins [17,18]. These post-translational modifications are termed S-nitrosylation and tyrosine nitration [19,20,21], which both shape cellular processes.

Drought or salt stress typically promotes massive transcriptional reprogramming, which initiates processes that protect plants from damage arising from these environmental stresses [22]. Several transcription factors (TFs), such as AP2/ERF, MYB, WRKY, NAC, bZIP, BHLH, HSF2, DREB, and CBF [23,24,25,26,27,28]), are known to regulate the response of plants to abiotic stress. These TFs orchestrate the transcriptional regulation of a plethora of downstream genes, which ultimately increase the tolerance of plants to single or multiple stresses (e.g., heat, drought, salt, cold, and/or oxidative stress).

Despite extensive research, thousands of Arabidopsis genes remain to be characterized. One such group is the “domain of unknown function” (DUF) genes that encode the large DUF protein family represented in the Pfam database (http://pfam.xfam.org/family) (accessed on 12 April 2021). In this context, the function of ~3600 DUFs is currently undefined [29,30].

Many DUFs possess high structural conservation; these genes are thought to play important roles in plant cellular and molecular processes. In contrast, several other DUFs are considered either less important or important only during specific conditions. For instance, DUF143, which has been studied in *Escherichia coli* and is commonly found in most bacteria and eukaryotic genomes, does not contribute to any form of change in the bacterial phenotype [31]. Studies have reported a potential role of DUF genes in plant abiotic stress conditions. For instance, when the TF *Triticum aestivum* Salt Response Gene (*TaSRG*) or *DUF622* were overexpressed in Arabidopsis, the resulting plants showed increased salt tolerance relative to that of wild-type (WT) Arabidopsis plants [32]. *AtRDUF1* and *AtRDUF2* (RING-DUF1117E3), which function as ubiquitin ligases, also function in response to abiotic stress, specifically in the regulation of ABA-related drought stress. Single and double loss-of-function mutations within these genes reduce tolerance to ABA-mediated drought stress [33]. However, most research on DUF genes and their role in abiotic stress has been conducted using rice plants. Recently, four rice (*Nipponbare*) *OsDUF866* family members (*OsDUF866.1* to *OsDUF866.4*) with three distinct motifs were examined under abiotic stress and ABA treatment conditions [34]. Differential expression of all these genes was observed under drought, salt, or heat conditions. Similarly, studies on *OsDUF872*, *OsDUF829*, and *OsDUF946* in rice [34,35,36,37] have demonstrated a potential role for these genes in the tolerance of rice to abiotic stress.

Previously, we identified a connection between DUF genes and NO following transcriptomic analysis of Arabidopsis leaves in response to NO accumulation [38,39]. Based on the highest fold change (57.29) we selected novel *DUF569* (AT1G69890) as a protein-coding gene for further characterization. The genomic context of the gene is located on chromosome 1 with the sequence information NC_003070.9 (26323086...26324946, complement). We identified the putative role of this gene under drought stress and examined its expression patterns under normal and drought stress conditions. Furthermore, DUF569 had interactive protein partners, such as E3 ubiquitin ligase and glycine reach proteins, which are reported to play roles in the drought stress response [40,41,42]. To the best of our knowledge, however, novel *DUF569* (AT1G69890) has yet to be functionally characterized. In the present study, we investigated *DUF569* (AT1G69890) and observed that its functional loss resulted in plants of greater stature with increased branching, higher seed yield, and longer siliques. We also examined the role of *DUF569* (AT1G69890) in drought stress, demonstrating that during the rosette leaf stage following three days of water withdrawal, *atduf569* plants typically showed severe wilting. Moreover, after rewatering, these plants failed to recover. Thus, our results suggest that DUF569 may play an essential role in controlling drought stress in Arabidopsis.

## 2. Materials and Methods

### 2.1. Experimental Design, Growth Conditions and Material

Seeds of the following Arabidopsis accessions were obtained from NASC (Nottingham Arabidopsis Stock Centre) (http://arabidopsis.info/) (accessed on 12 April 2021): WT-Col 0 (control), *atnced3* (drought-sensitive), *atcat2* (oxidative stress-responsive), and the DUF569 (*At1g69890*) knockout (KO) mutant *atduf569*. We replicated drought stress using the water withdrawal method with minor modifications [43]. Arabidopsis mutant and WT plants were of the Col-0 genetic background. All plants were grown at 22 ± 2 °C under long day photoperiod (with a 16:8 h light: dark). Plants were genotyped using T-DNA left border primers and gene-specific primers to identify homozygous mutant plants as described previously [39]. Arabidopsis seeds were germinated and grown on a peat moss substrate mixture with perlite and vermiculite (in a 1:1 ratio). To maintain appropriate soil moisture in control plants, water was provided by irrigation regularly.

### 2.2. Seed Sterilization

Seeds of all the WT and mutant genotypes were surface-sterilized for 5 min in a 50% bleach solution containing 0.1% Triton X-100 (Sigma Aldrich, Spruce Street, Saint Louis, MO 63103, USA). Seeds were subsequently rinsed with sterile distilled water 3–5 times and then incubated at 4 °C for 24 h to obtain uniform germination.

### 2.3. Water Withdrawal

For drought stress experiments, 4-week-old plants were used. WT Col-0, *atgsnor1-3*, *atnced3*, *atcat2*, and the *atduf569* KO mutant plants were subjected to drought stress as described by [44]). Initially, all the plants were irrigated with an equal amount of water; after complete absorption of this water, the plants were transferred to dry tray bases. Plants were subjected to drought for 10 days, initial data was recorded at the 7-day stage, and final evaluation was carried out at 10 days, after which the plants were rewatered. Their rate of recovery was then calculated 24 h after rewatering (the entire process is displayed in Appendix A). Other hand, WT control plants were watered frequently. Electrolyte leakage and transcript level of genes involved in drought stress were measured. All the genotypes were assessed for their phenotypic response to drought stress. In addition, soil moisture content was monitored throughout the experiment. Briefly, to assess water loss in the soil, 3 × 5-well pots for each genotype were weighed in triplicate regularly and the percentage water loss was calculated as a percentage of the actual weight loss from the initial weight of the saturated soil (considered “100% soil moisture”). Leaf samples were collected when symptoms of loss of turgidity and wilting began. Leaves were examined for gene expression, proline content, antioxidant enzyme activity (non-enzymatic and enzymatic), lipid peroxidation, chlorophyll, carotenoid content, and electrolyte leakage. The phenotypic response was recorded 10 days after drought, whereas the recovery rate was calculated 24 h after plants were rewatered. Soil moisture content, low water content, and the dynamic response of leaf transpiration (leaf wilting) to decreasing soil water content were monitored regularly.

### 2.4. Measurement of Electrolyte Leakage

To measure drought-induced cell membrane damage or injury, electrolyte leakage was analyzed using the method described by [45] with slight modifications. At 7 days after being subjected to drought, leaf samples were harvested from control plants and mutant plants. Uniform leaf discs (1 cm in diameter) were collected in triplicate (from different leaves) to comprise a replicate. The leaf discs were then rinsed with deionized distilled H_2_O to remove any surface electrolytes. Washed leaf discs were transferred into glass tubes with 5 mL of deionized H_2_O, and the glass tubes were maintained at ambient temperature for 48 h with continuous gentle shaking. Subsequently, the electrolyte leakage of each sample was measured at different time points and data were recorded over time for 48 h (at 0, 3, 6, 12, 24, and 48 h) using a portable conductivity meter (Huriba Twin Cond B-173, Japan). Electrolyte leakage was measured using Formula (1) as follows:EL (%) = (EL1/EL2) × 100,(1)
where EL (%) is the percentage of electrolyte leakage, EL1 is electrolyte leakage 1 (initial conductivity), and EL2 is electrolyte leakage 2 (for which samples were autoclaved at 120 °C for 20 min, cooled at ambient temperature, and then electrolyte leakage was measured again).

### 2.5. Quantitative Real-Time PCR Analysis

Leaf tissue samples from the drought and control conditions were collected at two time points (3 and 7 days). Total RNA was extracted using the method described [39]. In brief, total RNA was extracted with Trizol (Invitrogen, Carlsbad, CA, USA), and the extracted RNA was used with reverse transcriptase kits (BIOFACT, Daejeon, Korea) to synthesize first-strand complementary DNA (cDNA) according to the kit manufacturer’s instructions. The synthesized cDNA was then used for quantitative real-time PCR (qRT-PCR)-based gene expression analysis. The total reaction volume was 20 µL, with the reaction mixture containing 2× Real-Time PCR Master Mix including SYBR Green I (Biofact, Daejeon, Korea) and 10 pmol of each primer, which was processed in a two-step PCR program. The program included initial denaturation at 95 °C for 15 min followed by denaturation at 95 °C for 15 s and annealing and extension at 60 °C for 30 s. Melting curves were assessed at 60–95 °C to verify the amplicon specificity of each primer pair. Actin was used as an internal reference gene (Appendix A). All reactions were performed using a PCREco real-time PCR system (Illumina, San Diego, CA, USA).

### 2.6. Lipid Peroxidation Measurements

Lipid peroxidation in Arabidopsis was measured spectrophotometrically as described by [46]. Briefly, around 100 mg of leaf tissue was ground in liquid nitrogen and homogenized with 10 mL 10 mM phosphate buffer at pH 7.0. Subsequently, 200 μL of 8.1% sodium dodecyl sulfate, 1.5 mL of 20% acetic acid (pH 3.5), and 1.5 mL of 0.81% thiobarbituric acid were added as an aqueous solution to the supernatant. The resulting reaction mixture was incubated in boiling water for 60 min, before being cooled at ambient temperature. After cooling, 5 mL of butanol: pyridine (15:1 *v*/*v*) solution was added. The upper layer of the reaction mixture was recovered, and the absorption of the resulting pink-colored sample was measured at 532 nm using a spectrophotometer. The lipid peroxidation concentration was measured as the amount of malondialdehyde per gram of tissue weight (nmol MDA/g wet weight).

### 2.7. Quantification of Total Protein

Total protein in both control and drought stress plants was estimated based on the spectrophotometric assay of Bradford (1976) with slight modifications. Briefly, leaf tissue (250 mg) was sampled and the supernatant extracted from this tissue, 10 μL of which was used in the Bradford assay with the addition of 190 μL of Bradford reagent. The absorbance was then measured at 595 nm (vs. the blank) using a spectrophotometer (Multiskan GO, Thermo Fischer Scientific, Vantaa, Finland). The protein concentration was expressed as µg g^−1^ fresh weight.

### 2.8. Quantification of Antioxidants

The activities of antioxidant enzymes catalase (CAT), polyphenol oxidase (PPO) peroxidases (PODs), and superoxide dismutase (SOD) were evaluated using the spectrophotometric method described by [47].

#### 2.8.1. CAT Activity

The CAT enzyme assay described by [47] was used with some modifications to determine CAT activity in Arabidopsis leaf tissue under drought stress. Briefly, 100 mg of leaf tissue from control and drought stress treated plants was ground and homogenized in 10 mL of 0.1 M potassium phosphate buffer (pH 6.8). Samples were centrifuged at 10,000 rpm. Then 100 μL of the supernatant was transferred to a fresh tube, and 100 μL of 0.2M H_2_O_2_ was added. Samples were incubated for 10 min, and the absorbance was recorded at 240 nm using a spectrophotometer (Multiskan GO, Thermo Fischer Scientific, Vantaa, Finland).

#### 2.8.2. PPO and POD Activities

PPO activity in leaf tissue was determined according to the method of [48] with minor modifications. Briefly, leaf (100 mg) tissue was homogenized with 10 mL 0.1 M potassium phosphate buffer (pH 6.8) and then centrifuged for 15 min at 5000 rpm and 4 °C. Then, 100 μL of the supernatant was transferred to another tube, and 50 μL of pyrogallol (50 μM) was added. This mixture was incubated at 25 °C for 5 min, and the enzymatic reaction was then stopped by adding 5% 1 N H_2_SO_4_ (*v*/*v*). Subsequently, the absorbance was measured at 420 nm using a spectrophotometer (Multiskan GO, Thermo Fischer Scientific, Vantaa, Finland), and the quantity of purpurogallin formed was estimated as a 0.1-unit increase in absorbance. To determine POD activity, the same procedure was followed using the same reagents, but 50 μL of H_2_O_2_ was also added to the extraction buffer. The resulting reaction mixture was analyzed as mentioned for PPO.

#### 2.8.3. SOD Activity

SOD-like enzyme activity was determined following the methods of [47,49] with some modifications. Briefly, a reaction assay mixture was prepared using 150 μL of Tris–HCl, 10 mM EDTA, 150 μL 50 mM Tris (pH 8.5), and 200 μL of 7.2 mM pyrogallol. This mixture was added to 200 μL of leaf tissue extract (obtained by grinding (100 mg) tissue in the 10 mL 0.1 M potassium phosphate buffer (pH 6.8)) and incubated at ambient temperature (22–25 °C) for 10 min in the dark. Subsequently, 50 µL of 1 N HCl was added to stop the reaction. The level of pyrogallol autoxidation inhibition was then measured at an absorbance of 420 nm using a microplate spectrophotometer (Multiskan GO, Thermo Fischer Scientific, Vantaa, Finland).

### 2.9. Total Flavonoid Content

The total flavonoid content of Arabidopsis leaf tissue was determined as reported by [49,50] with minor modifications. Leaf (100 mg) tissue extract (300 µL) was mixed with an equal volume of double-distilled H_2_O, and then 30 µL of 5% NaNO_2_ was added to produce a reaction mixture. This was incubated for 5 min at ambient temperature before 60 µL of 10% AlCl_3_ was added. This mixture was again incubated for 5 min at ambient temperature and then 200 µL of 1 M NaOH was added. The resulting mixture was used to measure spectrophotometric absorbance at 500 nm using a microplate spectrophotometer (Multiskan GO Thermo Fischer Scientific, Vantaa, Finland). For calibration curve analysis, quercetin was used as a standard. The total flavonoid content was determined as μg quercetin equivalent g^−1^ sample. The assay was performed with three replications.

### 2.10. Amino Acid Content

Hydrolyzed amino acid extraction and quantification were performed as described by [48]. Specifically, 100 mg of leaf tissue was hydrolyzed with 1 mL of 6 N HCl for 24 h at 110 °C and then vacuum filtrated at 80 °C. Thereafter, the residue was dissolved in 2 mL of deionized H_2_O, and the previous step was repeated twice. The resulting residue was then dissolved in 1 mL of 0.02 N HCl solution, mixed properly, and filtered through a 0.45-µm cellulose acetate membrane. The amino acid composition was determined using an automatic amino acid analyzer (L-8900 Hitachi, Tokyo, Japan). Amino acid standards (Wako Pure Chemical Industries, Ltd., Osaka, Japan) were used to quantify the amino acids in tested samples. All the assays were conducted with replication. The quantified amino acids were expressed as µg sample^−1^ (Appendix A).

### 2.11. Chlorophyll and Total Carotenoid Content

To determine total chlorophyll in the samples, a previously reported method was used [51]. Specifically, leaf tissue (500 mg) was ground, and then the photosynthetic pigment was harvested using 80% acetone. Chlorophyll *a* and *b* were determined according to the methods of [51] using a spectrophotometer (Multiskan GO, Thermo Fischer Scientific, Vantaa, Finland) with absorbance at 663 nm. Similarly, total carotenoid content was determined according to the method of [52], with the extracted supernatant examined at an absorbance of 480 nm via a spectrophotometer.

Chlorophyll *a* and *b*, and carotenoid content were calculated using the following formula:Chlorophyll *a* (mg g^−1^ FW) = [{(12.7 × A663) − (2.69 × A645)}/1000 × FW] × V,(2)
Chlorophyll *b* (mg g^−1^ FW) = [{(22.9 × A645) − (4.68 × A663)}/1000 × FW] × V,(3)
Carotenoids (µg g^−1^ FW) = A480 + (0.638 × A663) − (0.638 × A645),(4)
where A is the absorbance at the respective wavelength, FW is the fresh weight, and V is the extraction volume.

### 2.12. Measurement of ABA Content

To quantify endogenous ABA, the method of [53] was followed. Specifically, leaf (50 mg) tissue was lyophilized and ground, and then a sample was extracted with 95% isopropanol,5% glacial acetic acid, and 20 ng of [(±)-3,5,5,7,7,7-d6]-ABA. Thereafter, samples were chromatographed with a 0.3 mL ABA standard (100 ng mL). The concentrated extracts were cleaned with 3 mL 1 M sodium hydroxide (NaOH) (pH 12.5) and chlorophyll was removed using dichloromethane (CH_2_Cl_2_). Ethanol (EtOH) was added, and then the supernatant was collected and concentrated. The residue was washed with phosphate buffer (pH 8), polyvinylpolypyrrolidone was added, and the mixture was centrifuged at 130 rpm for 40 min. Afterward, the mixture was filtrated and partitioned using solvent–solvent extraction with ethyl acetate (EtOAc). The residue was then evaporated and further eluted by 10 mL diethyl ether:methanol (3:2, *v*/*v*) and 10 mL CH_2_Cl_2_. ABA extracts were withdrawn, dried, and methylated using a GC–MS/SIM instrument (6890N Network GC System) and a 5973 network mass selective detector (Agilent Technologies, Palo Alto, CA, USA). Lab-based data system software was used to quantify ABA content.

### 2.13. Statistical Analysis

All the experiments were performed independently and repeated three times. The data were measured in triplicate and statistically analyzed using GraphPad Prism software (Version 7.00, 1992–2016 GraphPad). The data were analyzed for standard error (±SE) and Student’s *t*-test to determine significant differences at 5% level of significance.

## 3. Results

### 3.1. Phenotypic Trait Characterization of the Atduf569 Plants

Before the drought stress experiment, we evaluated the *atduf569* KO mutant line phenotypes under normal conditions until the harvest of seeds. Eight-week-old *atduf569* plants had a significantly higher number of rosette branches than WT Col-0 plants (Figure 1D). Similarly, the total number of pod and tillers were significantly higher than WT Col-0 plants (Figure 1F,H). Likewise, other plant attributes such as plant height, total seed weight, pedicel length, and silique length of *atduf569* plants all showed an increase compared with WT Col-0 plants (Figure 1C,E,G,I).

### 3.2. Impact of Drought on Plant Phenotype and Survival

To examine the possible role of DUF569 in the drought stress response, we exposed 4-week-old WT-Col-0, *atnced3* (drought-sensitive) plants lacking the 9-cis-epoxycarotenoid dioxygenase (NCED3) gene that is essential for ABA biosynthesis during drought stress [54], *atcat-2* (oxidative stress-responsive) lacking the cationic amino acid transporters (*AtCAT2*) gene [11], and *atduf569* ecotypes to drought stress by withholding water for 10 days, as shown in Appendix A. The mutant plants exhibited high sensitivity towards drought stress in the early days of drought exposure; within the third day, they showed severe effects that included the wilting of leaves, which were more pronounced than the effects seen in the WT and drought-sensitive *atnced3* plants (Figure 2). After rewatering following 10 days of drought, the survival rate (24 h after rewatering) was substantially lower in *atduf569* plants relative to WT plants.

The soil moisture content of *atduf569* plants was significantly reduced relative to the WT line after 3 days of drought treatment; this metric then reduced further from the third to the tenth day (Figure 3A). The mutant plants continued to be highly sensitive to drought as shown by the monitoring of soil moisture content by loss of weight recorded during the 10-day exposure period (Figure 3A). In addition, the highest electrolyte leakage was observed in *atcat2* leaf tissue followed by *atduf569*, WT Col-0, *atnced3*, and *atgsnor1-3* plants, compromised in the function of GSNOR, which turns over S-nitrosoglutathione (GSNO) [55,56]. This indicates that the *atduf569* plants might have experienced more severe cellular membrane damage that resulted in increased electrolyte leakage (Figure 3B).

### 3.3. Antioxidant Activity and Total Protein under Drought Stress

To assess drought-induced oxidative stress, we determined the levels of the antioxidant enzymes PPO, POD, CAT, and SOD in Arabidopsis plants. PPO enzyme levels did not differ significantly among the tested plant genotypes (Figure 4A). However, CAT (Figure 4B) and SOD (Figure 4D) enzyme activity were significantly higher in *atduf569* drought-stressed plants compared with WT Col-0. On the other hand, POD enzyme activity was significantly reduced in *atduf569* drought-stressed plants compared with WT Col-0 (Figure 4C). In addition, the mutant *atduf569* line showed an increased level of total protein under drought stress compared with WT Col-0 plants (Figure 4E). Although CAT and SOD levels were higher in *atduf569* plants under drought stress, this increase in antioxidants may not be sufficient to counteract the ROS-mediated oxidative stress. Hence, the *atduf569* plants still showed high sensitivity towards drought stress.

Lipid peroxidation is one of the defense mechanisms of the plants exposed to drought stress. As electrolyte leakage is an indicator of membrane damage, MDA content is an indicator of cell membrane integrity and peroxidation level. To further support this, we examined the MDA content to assess the oxidative damage; in our case, the MDA content was significantly higher in *atduf569* plants compared to the WT Col-0 (Figure 5F). This level of MDA content suggests increased peroxidation and more membrane permeability, which causes more susceptibility for drought stress than those which produce less MDA.

### 3.4. Evaluation of Genes Involved in Nitrate Reductase, Strigolactone Signaling, Cellular Redox, and Iron Tolerance

In plants, regulation of plant growth and development is typically modulated by nitrate reduction and ammonia assimilation [57,58]. In Arabidopsis, nitrate reductase (NR) is encoded by two genes, *NIA1* and *NIA2* [59]. It is considered that NR activity is an indicator of stress tolerance in plants [60,61]. Hence, the gene expression pattern of *AtNIA1* and *AtNIA2* was examined in Arabidopsis plants under drought stress.

The *atduf569* KO mutant showed the highest *AtNIA1* expression levels at 3 days of drought stress compared with all other genotypes and with the control plants. However, *AtNIA1* expression levels had rapidly decreased at 7 days of drought stress compared with other genotypes, such as Col-0, *atgsnor1-3*, and *atnced3* (Figure 5A). A similar expression pattern was observed for *AtNIA2* (Figure 5B). In contrast, the gene expression of *AtGSNOR* was significantly higher in *atduf569* at 7 days of drought stress compared with control plants. Moreover, at 7 days of drought stress, *AtGSNOR* expression was higher in *atnced3 and atduf569* than all other genotypes (Figure 5C). In plants, *AtHK3* is a key regulator and responsible for ABA signaling and antioxidant defense [62]. Thus, we examined the *AtHK3* expression, which increased significantly in *atduf569* plants at the third and seventh days of drought stress relative to the control plant, and similar expression patterns were observed for WT Col-0, *atgsnor1-3*, *atnced3*, and *atcat2* (Figure 6D). In Arabidopsis, the more axillary growth 2 (*MAX2*) gene is functional in regulating strigolactone synthesis, and also plays a regulatory role in response to drought stress [34,63,64]. Therefore, we examined *AtMAX2* expression, which was also significantly higher in *atduf569* plants than in control plants at 3 and 7 days of drought stress. Interestingly, the *atgsnor1-3* expression pattern at 7 days was highest among all genotypes (Figure 5E). *AtRGA1* is a Repressor Gibberellin-Insensitive (RGA) member of the GRAS family, which is responsible for inhibiting plant growth and development [65]. Therefore, we examined *AtRGA1* expression, which was significantly increased in *atduf569* plants at the third and seventh days of drought stress relative to the control plants, with similar expression patterns exhibited by the other genotypes. Among all genotypes, the WT Col-0 line showed the highest expression of *AtRGA1* on the third day of drought stress (Figure 5F). Thus, the expression of genes involved in the nitrate reductase pathway, ABA signaling, strigolactone signaling pathway, cellular redox control and iron tolerance, all genes involved in the positive regulation of drought stress [63,64,66], was diverse in *atduf569* plants.

### 3.5. Expression Analysis of ABA Synthesis and Drought Signaling Genes

The expression of drought-responsive genes was also determined in control and drought-stressed plants. Typically, in plants, TFs, such as dehydration responsive element binding (DREB), are responsible for improving stress tolerance and are involved in regulating the expression of several stress-inducible genes in ABA and drought signaling pathways. Therefore, we examined *AtDREB1*, *AtDREB2*, *AtAPX*, *AtABI1*, *AtABA2*, and *AtABA3* gene expression. *AtDREB1* was significantly higher in the *atduf569* KO mutant than in the drought-sensitive mutant *atnced3* as well as the other genotypes tested. By the seventh day of drought, *AtDREB1* expression was significantly reduced compared with 3-day stress levels, but it remained higher in *atduf569* plants than in WT Col-0 plants. However, levels were lower than in *atnced3* and *atcat2* plants (Figure 6A). A similar expression pattern was observed for *AtDREB2* expression in *atduf569*, i.e., significantly higher expression at day three of drought stress relative to the other plant lines tested (Figure 6B). In contrast, the expression of *AtAPX* was significantly lower in *atduf569* plants than in control and *atnced3* drought-stressed plants at the third and seventh days of stress (Figure 6C). The expression of *ABI1* in *atduf569* did not differ significantly from the other genotypes at 3 days of drought stress, but its expression was significantly higher than control and all other treatments at the seventh day of stress (Figure 6E). At 3 days of drought stress, *ABA2* expression in *atduf569* was significantly higher than in the control and *atnced3* drought-stressed plants. However, *ABA2* expression was significantly reduced on the seventh day of stress (Figure 6D). In contrast, *ABA3* expression in *atduf569* plants at 3 days of stress was significantly lower than expression in the control plant. However, the expression of this gene significantly increased on the seventh day of drought stress (Figure 6F).

### 3.6. Total Polyphenol and Flavonoid Content

Plants exposed to abiotic stress produce polyphenols and flavonoids, which function as antioxidants, thereby inhibiting ROS formation in the plant cell.

We investigated the flavonoid content, which was significantly reduced in drought-exposed *atduf569* plants compared with WT Col-0 (Figure 7A). In addition, endogenous polyphenol content of the *atduf569* line was significantly higher than WT Col-0 drought-exposed plants, whereas well-watered plants showed a non-significant change (Figure 7B). The analysis of total flavonoid suggested that endogenous secondary metabolite accumulation in *atduf569* plants was inhibited by drought stress compared to well-watered plants, whereas drought stress induced the endogenous polyphenol level but alone may not have been adequate to protect the plant from drought. Thus, the loss-of-function mutant differentially regulated the antioxidant level in response to drought. Hence, these results confirm that *DUF569* plays an important role in enhancing secondary metabolite function and mitigating drought stress in Arabidopsis.

### 3.7. Impact of Drought Stress on Leaf Chlorophyll and Carotenoid Content

Plants under abiotic stress alter the pigmentation. Typically, in the plant, pigments such as chlorophyll and carotenoids have several functions, including detoxification of plants from the ROS, as well as being involved directly in photosynthesis and oxidative stress defense mechanisms [67].

Thus, we examined total carotenoid (Figure 8A) and chlorophyll (Figure 8B–D) content, which was significantly reduced in drought-stressed *atduf569* mutant plants compared with control plants. Specifically, chlorophyll *a* (Figure 8B) and *b* (Figure 8C), and total chlorophyll (*a* + *b*) (Figure 8D) were reduced significantly in *atduf569* mutants exposed to drought stress at the rosette stage.

### 3.8. Effects of Drought on Endogenous ABA Content

Plant phytohormone ABA is an endogenous messenger that plays a key role in a plant’s adaption to diverse stresses (drought, salinity, cold, and other abiotic stresses). Drought stress typically influences the ABA levels and highly increases endogenously.

To investigate this possibility, we determined the endogenous ABA content. We found that *atduf569* plants exposed to drought stress exhibited a significant reduction in ABA content (Figure 9). Drought stress, therefore, causes inhibition of ABA content in the *atduf569* line, which likely led to its sensitivity towards drought stress at the rosette stage.

## 4. Discussion

In the current work, we aimed to determine the putative functional role of the NO-induced *DUF569* (AT1G69890) gene in the response of Arabidopsis to drought stress. Drought can initiate excessive production of ROS and RNS in plants, which in turn damages cellular membranes and leads to oxidative stress [15,68]. Such oxidative stress can severely injure cellular components, resulting in cellular imbalance leading to cell death [69]. Plants under drought conditions can undergo diverse morphological, physiological, and developmental changes that lead to altered photosynthesis, production of antioxidants, and expression of genes involved in metabolism, defense, and cellular functions [70]. In the present study, we evaluated the Arabidopsis *atduf569* knockout mutant under drought conditions to elucidate the putative function of the *DUF569* gene in abiotic stress. This mutant showed high sensitivity towards drought stress within 3 days of exposure, e.g., severely wilting leaves (to a greater extent than the drought-sensitive genotype *atnced3*).

To determine the underlining molecular mechanisms, we also determined key antioxidant enzyme activities during drought stress. Several studies have shown that antioxidants protect plants from oxidative or osmotic stress caused by drought [69], during which the antioxidant system is activated, and the levels of antioxidant enzymes such as PPO, POD, SOD, and CAT increase [71]. We found that the *atduf569* mutant showed a significant increase in CAT activity under drought stress, a less pronounced increase in SOD function, and a significant reduction in POD enzyme activity. These results suggest that the overall antioxidant activity in the *atduf569* plant was not elevated substantially under drought stress, which might explain why the plant is sensitive to drought. Typically, drought stress negatively affects photosynthesis. This in turn reduces the synthesis of chlorophyll *a* + *b* and the amount of chlorophyll *a* + *b* binding proteins. These changes severely affect photosystem II and reduce levels of light-harvesting pigment [72].. Changes in photosynthetic activity can be determined by the levels of chlorophylls and carotenoids in plants [73]. Thus, we examined total chlorophyll and carotenoid content in the *DUF569* loss-of-function mutant. The content of chlorophylls (*a*, *b*, and *a* + *b*) and carotenoids was significantly lower in drought-stressed *atduf569* mutant plants at the rosette stage. This indicated that *AtDUF569* linked to the photosynthesis pathway could regulate the photosynthetic process in Arabidopsis plants. Moreover, a reduced level of chlorophyll content could lower the photosynthetic rate, which might alter plant growth under abiotic stress.

ABA is a well-studied phytohormone that plays an important role in regulating responses to drought stress through the mediation of stomatal movement and transpiration rate [74]. Studies have also shown that exogenous application of ABA or overexpression of ABA-related genes enhances the tolerance of plants to drought stress [75]. We determined the ABA content in *atduf569* plants and found that it was significantly reduced under drought stress. This suggests that the sensitivity of *atduf569* plants to drought may be due to an inability to produce adequate amounts of ABA. This supports the conclusion that *DUF569* plays a part in regulating the ABA biosynthesis pathway in Arabidopsis. Indeed, ABA-deficient mutants have previously shown sensitivity to drought stress [76,77]. We further tested this possibility by measuring the expression of ABA biosynthesis and drought signaling-related genes, such as *AtABA2*, *AtABA3*, *AtABI1*, *AtDREB1*, *AtDREB2*, and *AtAPX2*. Indeed, we found that the expression of ABA signaling-related genes (*AtABI1*, *AtDREB1*, and *AtDREB2*) was significantly higher in the *atduf569* mutant on the third day of drought stress. Hence, our results indicate that *AtDUF569* may be a positive regulator of ABA biosynthesis (directly or indirectly), as the KO mutant is drought sensitive. Still, the gene expression of *AtABA2*, *AtABA3*, *AtABI2*, *AtDREB1*, *AtDREB2*, and *AtAPX* was significantly higher in the KO mutant. This may be explained by the fact that these genes either function downstream of ABA biosynthesis or function in an ABA-independent manner. When induced by environmental cues such as drought and salt stress, the *AtABI2* (which functions downstream of ABA biosynthesis) is involved in Ca^2+^-dependent ABA signaling. Moreover, this protein works in conjunction with other proteins such as CBL/CIPK, CDPK, and other members of the ABI family to promote stress tolerance via regulation of stomatal movement and maintenance of growth and development [78]. The DREB proteins on the other hand are drought-specific but they are ABA-independent. DREB1 and DREB2 are members of a large AP2/ERF transcription factor family that play important roles in abiotic stress responses. The DREB proteins are further sub-divided into two sub-groups (A1 and A2). Following the perception of drought and salt stress signals, DREB genes activate an ABA-independent pathway to regulate the expression of downstream genes that are involved in stress mitigation [79,80]. Ascorbate peroxidase (APX) is a plant antioxidant enzyme involved in the scavenging of excess hydrogen peroxide under stress conditions. Therefore, APX functions downstream of ABA biosynthesis.

Only the *ABA2* and *ABA3* genes are involved in ABA biosynthesis as they function upstream of ABA biosynthesis. The conversion of xanthoxin to abscisic acid aldehyde is catalyzed by *AtABA2* in Arabidopsis. The expression of these genes in the *atduf569* KO line, although higher than in the WT plants, appears to be insufficient for promoting higher ABA production in this line, as indicated by ABA measurement results. It is clear that a consistent or increased ABA content requires multiple gene functions, and high expression of *ABA2* and *ABA3* alone are not enough to guarantee an increase in ABA content in the *atduf569* mutant following drought stress. In addition, *AtHK3* expression levels were significantly increased in *atduf569* plants under drought stress at the third and seventh days (Figure 5D). Consistent with these findings, *AtHK3* is known to be a crucial regulator of ABA signaling and antioxidant defense [62].

We also examined secondary metabolites, such as flavonoids, polyphenols, and carotenoids, as they are known to increase in plants under abiotic stress conditions [81]. However, flavonoids, polyphenols, and carotenoids were all significantly reduced in drought-stressed *atduf569* plants.

Previously, it was reported that endogenous levels of NO are induced under osmotic stress [82]. Endogenous NO biosynthesis involves different enzymes, one of which is nitrate reductase encoded by *NIA* genes [83,84]. *NIA1* and *NIA2* were found to contribute 10% and 90% of total nitrate reductase activity, respectively [85,86]. To assess the generation of NO under drought stress in *atduf569* plants, we examined the expression of these genes. We found that the *atduf569* plants showed high levels of *AtNIA1* and *AtNIA2* expression at 3 days of drought stress, which suggests that NO production may have peaked on the third day of drought treatment. Previously, studies have reported that *GSNOR* controls RNS levels under abiotic stress [15], NO and S-nitrosothiol (SNO) content are increased under drought or waterlogging and hypoxia, while *GSNOR* activity is reduced and protein tyrosine nitration increased [87]. However, our data show that *AtGSNOR* was expressed at significantly higher levels in *atduf569* plants on the third day of drought stress relative to WT Col-0 plants, while expression was further increased at the seventh day of stress. Thus, *AtGSNOR* expression in *atduf569* plants is higher than all the other tested Arabidopsis genotypes.

Recently, the Arabidopsis *MAX2* (more axillary growth 2) gene was found not only to be a central regulator of strigolactone synthesis but also to play a regulatory role in the response to drought stress [34,63,64]. We found that the expression level of *AtMAX2* in *atduf569* plants was higher than in WT Col-0 plants on the third and seventh days of drought. Interestingly, however, the expression of *AtMAX2* on 7 days of the drought was highest in the *atgsnor1-3* genotype.

Various reports have shown that GRAS TF family genes, such as *HcSCL13*, *ZmSCL7*, *AtRGA*, and *AtGAI*, participate in abiotic stress responses in higher plants [88].. We explored the expression of *AtRGA1* under drought stress. The accumulation of this transcript was significantly higher in *atduf569* plants on the third and seventh days of drought relative to expression in WT Col-0 plants, and a similar pattern was observed for other tested Arabidopsis genotypes. Among all genotypes, however, the WT Col-0 plants showed the highest expression level of *AtRGA1* on the third day of drought treatment. A schematic illustration of the proposed function of *DUF569* in Arabidopsis and its relationship with ROS, NO, and other genes described above is given in Figure 10.

## 5. Conclusions

Our results demonstrate that *AtDUF569* is a positive regulator of the functional response to drought stress in Arabidopsis. This study improves our understanding of the role of *AtDUF569* in the response of Arabidopsis to abiotic stress and provides insights towards the elucidation of the molecular mechanism underlying the drought-stress response in plants. However, further research will be required to validate the role of *AtDUF569* in drought stress regulation across plant species.

## Figures and Tables

**Figure 1 ijms-22-05316-f001:**
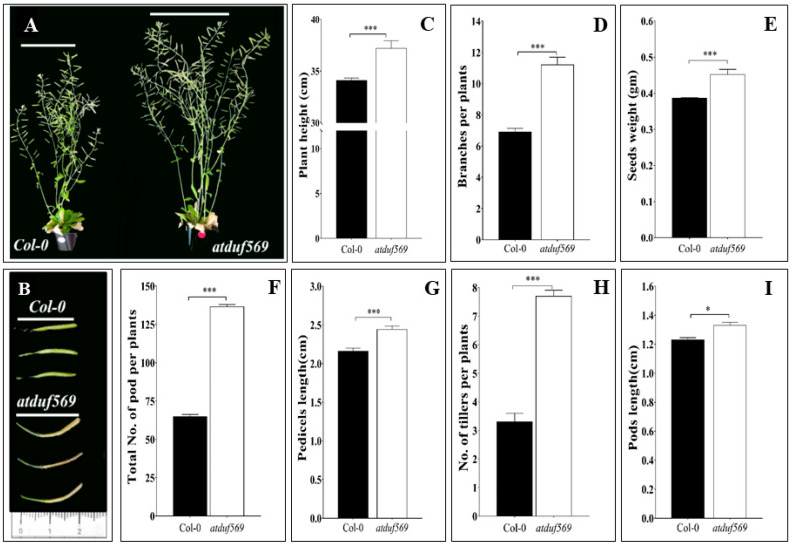
Phenotypic evaluation of the Arabidopsis *atduf569* mutant and WT Col-0 plants. (**A**) Phenotypic appearance of plants at the silique stage. (**B**) Pod length in the two genotypes. (**C**) Plant height of 8-week-old plants. (**D**) The number of branches per plant. (**E**) Seed weight measured after the final seed harvest. (**F**) The total number of pods of 8-week-old plants. (**G**) Pedicel length recorded in 8-week-old plants. (**H**) Tiller number in the two genotypes. (**I**) Pod length measured at the 8-week stage. Data represent means of values obtained from experiments performed in triplicate. Error bars represent standard deviation. Means were analyzed for significant differences using Student’s *t*-test. Asterisks * indicate significant differences at 5% level of significance (*** *p* < 0.001, * *p* < 0.05).

**Figure 2 ijms-22-05316-f002:**
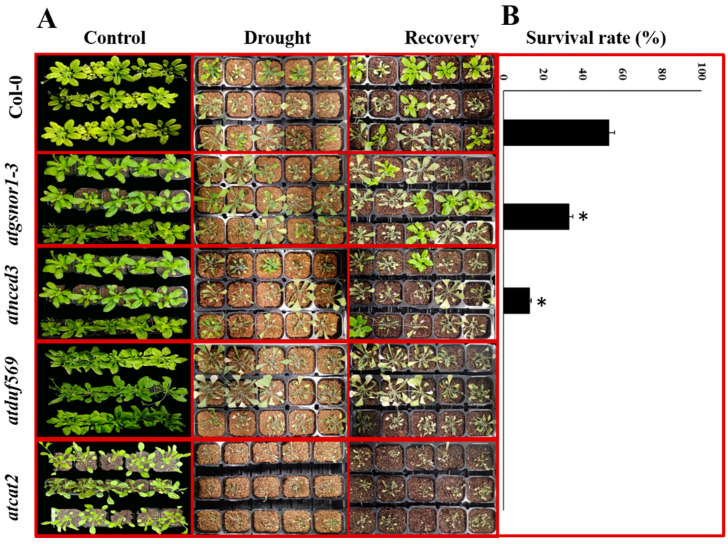
The phenotypes of Arabidopsis plants (WT Col-0, *atgsnor1-3*, *atnced3*, *atduf569*, and *atcat2*) in normal, drought, and recovery conditions. (**A**) Starting top to bottom with the left side (control) are *Col-0* (WT), *atgsnor1-3* (drought tolerant mutant), *atduf569* (KO mutant), *atnced3* (drought-sensitive mutant), *atcat2* (oxidative stress-responsive mutant). Next, the second (middle) part designated plants exposed to drought stress. The third (right) side part exhibiting plants of the designated genotypes after recovery from water withholding. (**B**) The survival rate after drought stress was initiated by withholding water and then rewatering (for 24 h after). Data represent means of values obtained from experiments performed in triplicate. Error bars represent standard deviation. Means were analyzed for significant differences using Student’s *t*-test. Asterisks * indicate significant differences at 5% level of significance (* *p* < 0.05).

**Figure 3 ijms-22-05316-f003:**
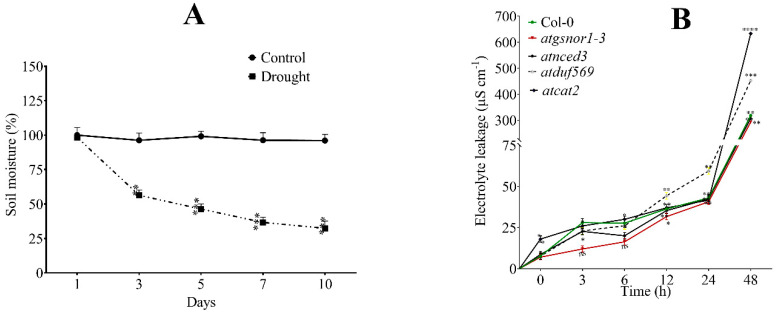
Soil moisture content and electrolyte leakage activity of the given Arabidopsis plants under drought stress. (**A**) The soil moisture content of the *atduf569* knockout mutant genotype under drought-stressed conditions by recording the loss of weight during the experiment. (**B**) Electrolyte leakage of Col-0, *atgsnor1-3*, *atnced3*, *atduf569*, and *atcat2* under the drought stress. Data represent means of values obtained from experiments performed in triplicate. Error bars represent standard deviation. Means were analyzed for significant differences using Student’s *t*-test. Asterisks * indicate significant differences at 5% level of significance (**** *p* < 0.0001, *** *p* < 0.001, ** *p* < 0.01, * *p* < 0.05).

**Figure 4 ijms-22-05316-f004:**
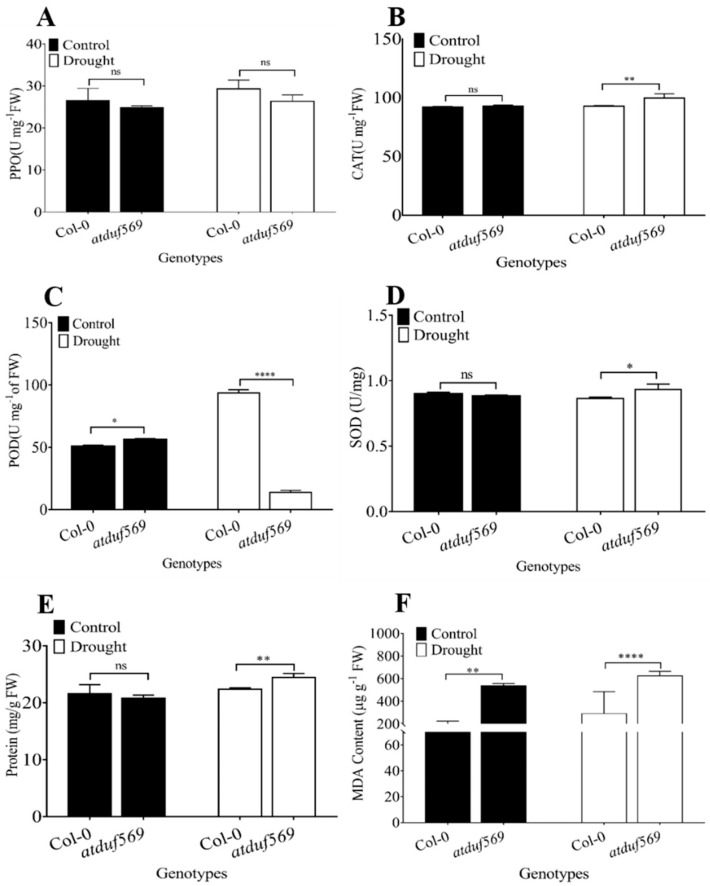
Antioxidant activity and total protein content of the given Arabidopsis plants under drought stress. Content of (**A**) polyphenol oxidase (PPO), (**B**) catalase (CAT), (**C**) peroxidase (POD), and (**D**) superoxide dismutase (SOD), and (**E**) total protein content and (**F**) MDA content. Data represent means of values obtained from experiments performed in triplicate. Error bars represent standard deviation. Means were analyzed for significant differences using Student’s *t*-test. Asterisks * indicate significant whereas “ns” indicate non-significant differences at 5% level of significance (**** *p* < 0.0001, ** *p* < 0.01, * *p* < 0.05).

**Figure 5 ijms-22-05316-f005:**
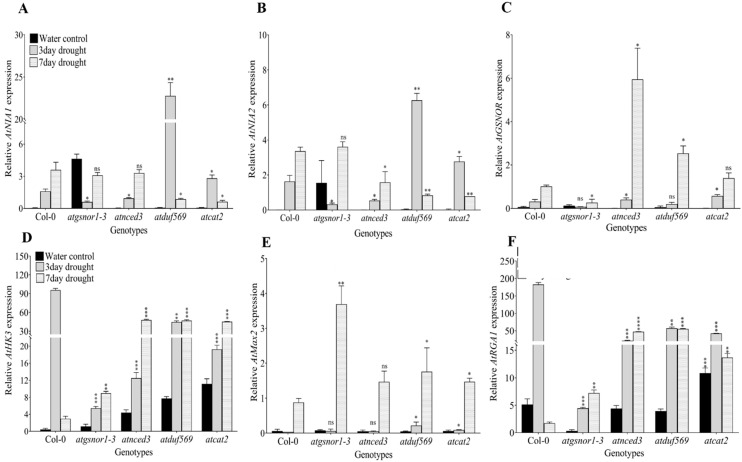
Expression analysis of genes involved in nitrate reductase, abscisic acid signaling, strigolactone signaling, cellular redox, and iron tolerance in the given Arabidopsis lines under drought stress. The gene expression levels of (**A**) *AtNIA1*, (**B**) *AtNIA2*, (**C**) *AtGSNOR*, (**D**) *AtHK3*, (**E**) *AtMAX2*, and (**F**) *AtRGA1* in various Arabidopsis genotypes. Data represent the means of values obtained from experiments performed in triplicate. Error bars represent standard deviation. Means were analyzed for significant differences using Student’s *t*-test. Asterisks (*) indicate significant whereas “ns” indicate non-significant differences at 5% level of significance (**** *p* < 0.0001, *** *p* < 0.001, ** *p* < 0.01, * *p* < 0.05).

**Figure 6 ijms-22-05316-f006:**
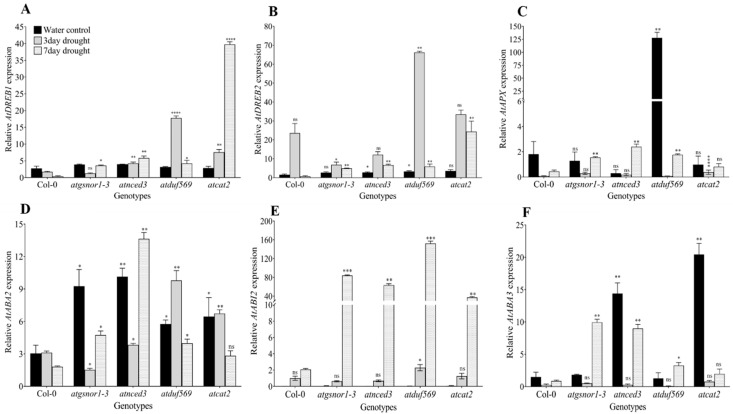
Expression analysis of genes related to abscisic acid synthesis and drought signaling in Arabidopsis plants under drought stress. Gene expression levels of (**A**) *AtDREB1*, (**B**) *AtDREB2*, (**C**) *AtAPX*, (**D**) *AtABA2*, (**E**) *AtABI2*, and (**F**) *AtABA3* in various Arabidopsis genotypes. Data represent means of values obtained from experiments performed in triplicate. Error bars represent standard deviation. Means were analyzed for significant differences using Student’s *t*-test. Asterisks * indicate significant whereas “ns” indicate non-significant differences at 5% level of significance (**** *p* < 0.0001, *** *p* < 0.001, ** *p* < 0.01, * *p* < 0.05).

**Figure 7 ijms-22-05316-f007:**
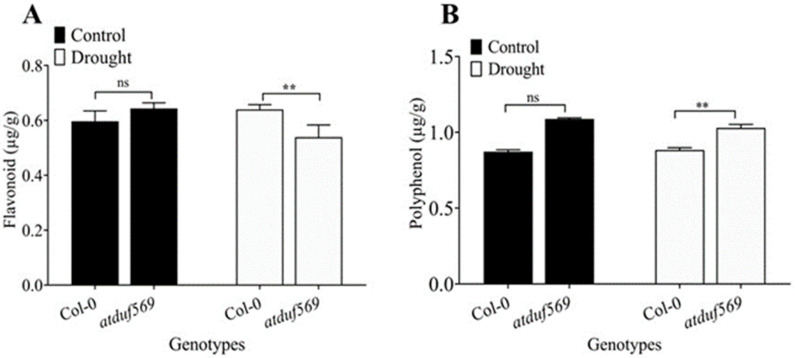
Total flavonoid and polyphenol content in the leaves of Arabidopsis plants under drought stress. (**A**) Total flavonoid and (**B**) total polyphenol contents of WT Col-0 and *atduf569* knockout mutant Arabidopsis genotypes under drought and control conditions. Data represent means of values obtained from experiments performed in triplicate. Error bars represent standard deviation. Means were analyzed for significant differences using Student’s *t*-test. Asterisks ** indicate significant whereas “ns” indicate non-significant differences at 5% level of significance (** *p* < 0.01).

**Figure 8 ijms-22-05316-f008:**
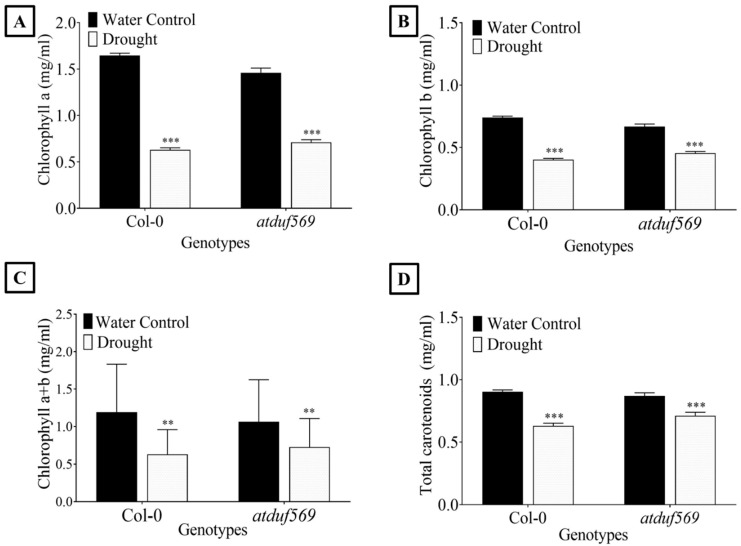
Total carotenoids and chlorophyll contents of WTCol-0 and *atduf569* knockout mutant Arabidopsis plants under drought stress. (**A**) Total carotenoids, (**B**) chlorophyll *a* pigment, (**C**) chlorophyll *b* pigment, and (**D**) chlorophyll *a* + *b* pigment in the two Arabidopsis genotypes under the drought and control conditions. Data represent means of values obtained from experiments performed in triplicate. Error bars represent standard deviation. Means were analyzed for significant differences using Student’s *t*-test. Asterisks * indicate significant whereas “ns” indicate non-significant differences at 5% level of significance (*** *p* < 0.001, ** *p* < 0.01).

**Figure 9 ijms-22-05316-f009:**
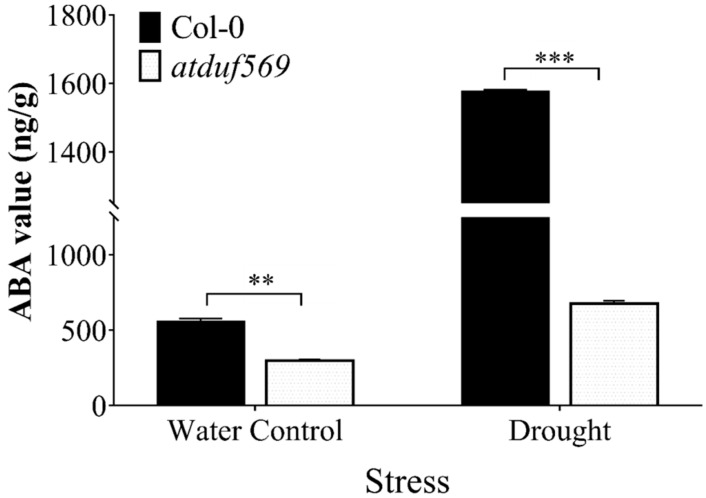
Change in abscisic acid ABA content of WTCol-0 and *atduf569* Arabidopsis plants under drought conditions. Data represent means of values obtained from experiments performed in triplicate. Error bars represent standard deviation. Means were analyzed for significant differences using Student’s *t*-test. Asterisks * indicate significant whereas “ns” indicate non-significant differences at 5% level of significance (*** *p* < 0.001, ** *p* < 0.01).

**Figure 10 ijms-22-05316-f010:**
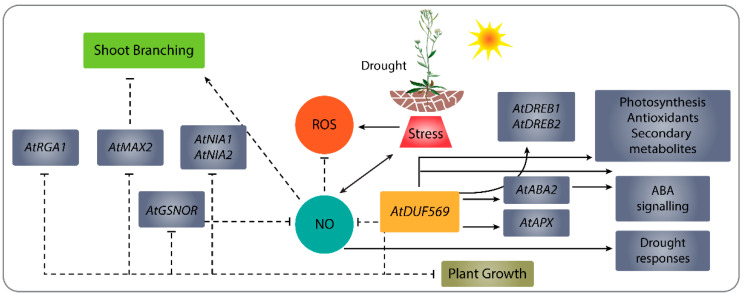
A simple schematic illustration of a proposed function of *DUF569* in *Arabidopsis thaliana* during drought stress.

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
