# Peer review of "A Novel DUF569 Gene Is a Positive Regulator of the Drought Stress Response in Arabidopsis"

_ijms, 2021, doi:10.3390/ijms22105316_

Round 1

Reviewer 1 Report

In this manuscript author studied a novel DUF569 gene that is a positive regulator of the drought stress response in Arabidopsis. The Arabidopsis loss-of-function mutant atduf569 showed significant sensitivity to drought stress, i.e., severe wilting at the rosette-leaf stage after water was withheld for 3 days. Importantly, the mutant plant did not recover after re-watering, unlike wild type plants. In addition, atduf569 plants showed significantly lower Abscisic acid accumulation under optimal and drought-stress conditions, as well as significantly higher electrolyte leakage, when compared with wild type Col-0 plants. Spectrophotometric analyses also indicated a significantly lower accumulation of polyphenols, flavonoids, carotenoids and chlorophylls in atduf569 mutant plants.

The manuscript is scientifically sound but have many grammatically errors and need some English polishing. I have also found some plagiarism stuff. Please clean it. For the betterment of the manuscript, I have few comments to make.

I would be better for authors to make some hypothetical last figure that predict the mechanism and function of the gene found in the study supplemented with the previously reported study for this particular gene.

Changes all wild type to wild-type.

Once wild-type (WT) written first time no need to wild-type or (WT) again together See at L104.

Change at

L106, L477 in the response to to in response to.

L132 demonstrating that, during to demonstrating that during.

L134 rewatering or re-watering. Other places it is written as re-watering at L168.

L249 what is CAT, PPO, POD, and SOD? Write down full form when first time introduced.

L341 What is NaOH and EtOA? Write down full form.

L360-361 Rearrange the line. Where?

L516, L522 than in the control to than in control.

L626 sensitive towards drought to sensitive to drought.

L670 plants at the third day to plants on the third day.

Remove from L691 to L749.

I have found plagiarism at L146-149, L170-171, L357-359, L600-602.

Author Response

uthor's Reply to the Review Report (Reviewer 1)

In this manuscript author studied a novel DUF569 gene that is a positive regulator of the drought stress response in Arabidopsis. The Arabidopsis loss-of-function mutant atduf569 showed significant sensitivity to drought stress, i.e., severe wilting at the rosette-leaf stage after water was withheld for 3 days. Importantly, the mutant plant did not recover after re-watering, unlike wild type plants. In addition, atduf569 plants showed significantly lower Abscisic acid accumulation under optimal and drought-stress conditions, as well as significantly higher electrolyte leakage, when compared with wild type Col-0 plants. Spectrophotometric analyses also indicated a significantly lower accumulation of polyphenols, flavonoids, carotenoids and chlorophylls in atduf569 mutant plants.

  1. The manuscript is scientifically sound but have many grammatically errors and need some English polishing. I have also found some plagiarism stuff. Please clean it. For the betterment of the manuscript, I have few comments to make.

Answer: We would like to thank the worthy reviewer for their time and constructive comments and suggestions to improve our manuscript. We have revised the MS, addressed all the comments, and made the suggested corrections. We hope that the revised manuscript would be suitable for publication now. All the corrections have been made with track change. The manuscript has been proof read and edited by native English language speakers.

  1. I would be better for authors to make some hypothetical last figure that predict the mechanism and function of the gene found in the study supplemented with the previously reported study for this particular gene.

Answer: We have made a figure representing a model mechanism for the function of DUF569 in Arabidopsis thaliana during drought stress. This figure has been added to the revised manuscript as Figure 10. The figure has been cited in the discussion section (Page,27 line 776-779).

  1. Changes all wild type to wild-type. Once wild-type (WT) written first time no need to wild-type or (WT) again together See at L104.

Answer. Thank you for the suggestion. Wild type has been replaced with wild-type at the first appearance. Later it has been mentioned as WT throughout the MS.

  1. L106, L477 in the response to to in response to.

Answer. Corrected (Page3 L113, Page 13 L524)

  1. L132 demonstrating that, during to demonstrating that during.

Answer. Corrected (Page 3, L143)

  1. L134 rewatering or re-watering. Other places it is written as re-watering at L168.

Answer. Corrected. It is “rewatering” throughout the manuscript now. (Page 3,4  L145,184)

  1. L249 what is CAT, PPO, POD, and SOD? Write down full form when first time introduced.

Answer. Done. Thanks (Page, 6 L265-266)

  1. L341 What is NaOH and EtOA? Write down full form.

Answer. Thanks. Full form written as sodium hydroxide (NaOH) and ethanol (EtOH) we corrected EtOA to EtOH. (Page 8 L362-364)

  1. L360-361 Rearrange the line. Where?

Answer. Corrected. (Page 10, L422-446)

  1. L516, L522 than in the control to than in control.

Answer. Corrected. (Page 13, L524, 527)

  1. L626 sensitive towards drought to sensitive to drought.

Answer. Corrected. (Page, 19 L683)

  1. L670 plants at the third day to plants on the third day.

Answer. Corrected. (Page 20, 21, L750-772)

  1. Remove from L691 to L749

Answer. Corrected. (Page21 L805-858)

  1. I have found plagiarism at L146-149, L170-171, L357-359, L600-602.

Answer: Sorry for inconvenience, this plagiarism is related to our own previous publication. However, we have rephrased the sentence in the revised MS. (Page, 4 L155-160, L186-190, L379-381, Page18, 656-659)

Reviewer 2 Report

Reviewer notes

General comments/modifications:

  1. The authors were able to address the objectives they laid out for this study. They presented sufficient data to support their hypothesis and presented them in a way all readers can understand the importance of their research.
  2. This may be beyond the scope of this current study, but I want to point out that it would greatly add value to the current study if the authors are able to quantify or measure beyond the transcript level of the important genes presented that characterized the function of DUF569. They did perform some enzymatic measurements, but maybe adding to the qPCR results of these important genes (NIA, ABA, MAX2, GSNOR) a protein interaction assay that can exhibit and measure activity will be ideal.
  3. I believe it will also be very strong if the authors can add an overexpressed mutant for DUF569 and test it for drought tolerance. Based on the current results of this study, it may show increased drought tolerance as Arabidopsis is innately drought sensitive, particularly Col-0.
  4. This can be a future study but finding and testing orthologs of DUF569 in other economically important crops will be vital to further increase the evidence of the importance of this gene in drought response of crops.

Specific comments/modifications:

Pg. 3, Line 132: Confirm use of “roseate” instead of using “rosette”

Pg. 3, Line 137: Suggestion to change to plural form “Materials and Methods”

Figure 1: Please include statistical significance markers with data legends in the bar graphs. It will create better illustration for the results presented.

Pg. 9-10, Lines 404-412: This paragraph is suppose to be the figure 2 description and yet it is formatted as part of the manuscript. This could be a journal formatting issue, but kindly correct it if possible. Please reference this comment for subsequent figure descriptions in the manuscript and modify accordingly.

Figure 3: Suggestion to show statistical significance markers on the line graphs to clearly illustrate statistical significance among the mutant lines and WT.

Pg. 16, Line 591: Confirm use of “roseate” instead of using “rosette”. Please check across the entire manuscript if necessary, modify.

Table 1: It could be another journal formatting issue, but kindly consider modifying this table to allow for better reading presentation (i.e., separate each data entries with lines/boxes).

Pg. 18, Line 625: Suggest replacing “…antioxidant activity in atduf569 plant is not increased…” to “…antioxidant activity in atduf569 plant was not elevated…”

Pg. 18, Lines 626-630: Suggestion to split into 2 or more sentences as the original sentence is a run-on/ compound sentence that is confusing for the readers.

Pg. 18, Lines 638-654: This discussion paragraph is confusing. The authors suggested that DUF569 could be involved in ABA biosynthesis regulation and thus the KO mutant resulted in drought sensitivity. However, the results they presented showed that there were higher ABA signaling-related gene transcripts which can show higher activity of ABA biosynthesis. I understand the results presented and it made sense, but kindly add a few more sentences to explain why the increased ABA signaling transcripts actually showed reduced ABA in the mutant plants to avoid this confusion for other readers.  

Author Response

Author's Reply to the Review Report (Reviewer 2)

Comments and Suggestions for Authors

Reviewer notes

General comments/modifications:

  1. The authors were able to address the objectives they laid out for this study. They presented sufficient data to support their hypothesis and presented them in a way all readers can understand the importance of their research.

Answer: We would like to thank the worthy reviewer for their time and constructive comments and suggestions to improve our manuscript. We have revised the MS, addressed all the comments, and made the suggested corrections. We hope that the revised manuscript would be suitable for publication now. All the corrections have been made with track change. The manuscript has been proof read and edited by native English language speakers.

  1. This may be beyond the scope of this current study, but I want to point out that it would greatly add value to the current study if the authors are able to quantify or measure beyond the transcript level of the important genes presented that characterized the function of DUF569. They did perform some enzymatic measurements, but maybe adding to the qPCR results of these important genes (NIA, ABA, MAX2, GSNOR) a protein interaction assay that can exhibit and measure activity will be ideal.

Answer: Thank you very much for your valuable suggestion, we agree that currently, it is beyond the scope for this study. However, we have shown the possible protein interaction for this DUF569 gene in our previous publication (Nabi RBS, Tayade R, Imran QM, Hussain A, Shahid M and Yun B-W (2020) Functional Insight of Nitric-Oxide Induced DUF Genes in Arabidopsis thaliana. Front. Plant Sci. 11:1041. doi: 10.3389/fpls.2020.01041). The insilico and experimental observations suggested that the AtDUF569 protein interacts with various proteins involved in cellular trafficking machinery and carbohydrate-binding and with glycine-rich proteins that participate in cellular stress responses and signaling. Particularly, we have searched for interactions between the CysNO-induced AtDUF569 (AT1G69890) and other proteins using the Search Tool for the Retrieval of Interacting Genes/Proteins (STRING; https://string-db.org/). We observed some interesting interactions between the DUF569 and other proteins, and we identified 10 predicted functional partners, which included the uncharacterized protein, AT3G49790, known as a carbohydrate-binding protein. Its function was described as ATP-binding, but the underlying biological mechanism remains unknown. The PHLOEM PROTEIN 2-LIKE A10 is another carbohydrate-binding protein located in the mitochondria and found in several plant species and at different growth stages. The CYB-1 and ACYB-2 protein is a potentially transmembrane ascorbate ferrireductase 2, which contains two-heme-cytochrome (1 and 2) and is involved in the catalyzation of ascorbate-dependent transmembrane ferric-chelate reduction. Thus, providing the same information for this manuscript would not be appropriate.  However, we highly appreciate the suggestion and we look forward to perform invitro and invivo protein interaction assays in our further study as we performed other experiments related to AtDUF569.

  1. I believe it will also be very strong if the authors can add an overexpressed mutant for DUF569 and test it for drought tolerance. Based on the current results of this study, it may show increased drought tolerance as Arabidopsis is innately drought sensitive, particularly Col-0.

Answer: 3. Thanks for the comment. The generation of over expression DUF569 lines is indeed a good idea. However, it is hard under our current circumstances. That is why we have used multiple other lines for comparison such as nced3, atgsnor1-3, cat2. These lines are already known for their response to drought stress via a variety of mechanisms. We hope that this comparison would support our findings at this stage while we work to generate an over expression DUF569 line for future use.

  1. This can be a future study but finding and testing orthologs of DUF569 in other economically important crops will be vital to further increase the evidence of the importance of this gene in drought response of crops.

Answer: 4. Thanks for the comment. Indeed this would provide pretty useful and valuable information. We are working on multiple plant species in our lab and we are trying to identify other homologs and orthologs of the gene in plants other than Arabidopsis and to determine how these findings in model plant systems could be applied to crop plants. Furthermore, insilico analysis involving DUF569 in our previously mentioned publication we determined ancestral/evolutionary relationship of Arabidopsis DUF genes. For this purpose, we used the top 10 upregulated and downregulated DEGs from our transcriptome data to identify DUF orthologs in agronomically important species such as rice, soybean, wheat, maize, and tomato. Interestingly, we obtained 151 hits with more than 90% sequence similarity. These 151 protein sequences from different plant species were then used to generate a phylogenic tree that showed a close evolutionary relationship between DUF genes from different plant species. So, we are looking forward to working with these orthologs in further detail.

Answer. Specific comments/modifications:

  1. 3, Line 132: Confirm use of “roseate” instead of using “rosette”

Answer. It is rosette. Corrected throughout the manuscript now.

  1. 3, Line 137: Suggestion to change to plural form “Materials and Methods”

Answer. Corrected (Page 3, L148)

  1. Figure 1: Please include statistical significance markers with data legends in the bar graphs. It will create better illustration for the results presented.

Answer. Statistical information has been added to figure legends throughout the MS (Figure 1 and all other Figures)

  1. 9-10, Lines 404-412: This paragraph is suppose to be the figure 2 description and yet it is formatted as part of the manuscript. This could be a journal formatting issue, but kindly correct it if possible. Please reference this comment for subsequent figure descriptions in the manuscript and modify accordingly.

Answer. Corrected. All figure legends double checked. (Page 10, 424-426, Page10,11 L446-449)

  1. Figure 3: Suggestion to show statistical significance markers on the line graphs to clearly illustrate statistical significance among the mutant lines and WT.

Answer. Statistical information has been added to all figure legends.

  1. 16, Line 591: Confirm use of “roseate” instead of using “rosette”. Please check across the entire manuscript if necessary, modify.

Answer. Its rosette. Corrected throughout the manuscript. (Page 17, L647)

  1. Table 1: It could be another journal formatting issue, but kindly consider modifying this table to allow for better reading presentation (i.e., separate each data entries with lines/boxes).

Answer. Done. Thanks (Page 18, L656-657)

  1. 18, Line 625: Suggest replacing “…antioxidant activity in atduf569plant is not increased…” to “…antioxidant activity in atduf569 plant was not elevated…”

Answer. Done. Thanks (Page 19, L683)

  1. 18, Lines 626-630: Suggestion to split into 2 or more sentences as the original sentence is a run-on/ compound sentence that is confusing for the readers.

Answer. Done. Thanks. (Page 19, L685-689)

  1. 18, Lines 638-654: This discussion paragraph is confusing. The authors suggested that DUF569 could be involved in ABA biosynthesis regulation and thus the KO mutant resulted in drought sensitivity. However, the results they presented showed that there were higher ABA signaling-related gene transcripts which can show higher activity of ABA biosynthesis. I understand the results presented and it made sense, but kindly add a few more sentences to explain why the increased ABA signaling transcripts actually showed reduced ABA in the mutant plants to avoid this confusion for other readers.

Answer. It’s a good observation. The above finding is explained below. (Page 19, L 712-738)

Our results do indicate that AtDUF569 may be involved in ABA biosynthesis (directly or indirectly) and the KO mutant is drought sensitive. Still, the expression of the genes (ABA2, ABA3, AtABI2, DREB1, DREB2 and APX) was higher in the KO mutant. This is because most of these genes function downstream of ABA biosynthesis. For example the ABI2, when induced by environmental cues such as drought and salt stress, the AtABI2 (which functions downstream of ABA biosynthesis) is involved in Ca2+ dependent ABA signaling where this protein works in conjunction with other proteins such as CBL/CIPK, CDPK and other members of the ABI family to promote stress tolerance via regulation of stomatal movement and maintenance of growth and development (Kumar et al., 2019). DREB proteins on the other hand are drought-specific but they are ABA-independent. The DREB1 and DREB2 are members of a large AP2/ERF transcription factor family that play important role in abiotic stress responses. The DREB proteins are further sub-divided into two sub-groups (A1 and A2). Following the perception of drought and salt stress signals, DREB genes activate an ABA-independent pathway to regulate the expression of downstream genes that are involved in stress mitigation mitigation (Shinozaki and Yamaguchi-Shinozaki, 2007; Ullah Jan et al., 2017).

The Ascorbate peroxidase (APX) is a plant antioxidant enzyme involved in scavenging of excess hydrogen peroxide under stress conditions. Therefore, APX functions downstream of ABA biosynthesis.

Only the ABA2 and ABA3 genes are involved in ABA biosynthesis as they function upstream of ABA biosynthesis. The conversion of xanthoxin to abscisic acid aldehyde is catalyzed by AtABA2 in Arabidopsis. The expression of these genes in atduf569 KO line is although higher than the WT plants, it appears to be insufficient for promoting higher ABA production in this line as indicated by ABA measurement results. It is clear that a consistent or increased ABA content requires multiple gene functions and high expression of ABA2 and ABA3 are not enough to guarantee an increase in ABA content in the atduf569 mutant.

This information has been added to the discussion section for a better explanation to the reader.

Round 2

Reviewer 1 Report

I am happy with the authors reply. Manuscript looks much better now. It can be accepted in its current format.

Author Response

Thank you very much for endorsing our MS. We highly appreciate the time and valuable suggestions of worthy reviewer.